# Antioxidant, Wound Healing Potential and In Silico Assessment of Naringin, Eicosane and Octacosane

**DOI:** 10.3390/molecules28031043

**Published:** 2023-01-20

**Authors:** Abbirami Balachandran, Sy Bing Choi, Morak-Młodawska Beata, Jeleń Małgorzata, Gabriele R. A. Froemming, Charlie A. Lavilla, Merell P. Billacura, Stepfanie N. Siyumbwa, Patrick N. Okechukwu

**Affiliations:** 1Department of Biotechnology, Faculty of Applied Sciences, UCSI University, Cheras 56000, Kuala Lumpur, Malaysia; 2Faculty of Pharmaceutical Sciences, Department of Organic Chemistry, Medical University of Sílesia, Jagiellonska, Str. 4, 41-200 Sosnowiec, Poland; 3Basic Medical Sciences, Faculty of Medicine and Health Sciences, Universiti Malaysia Sarawak (UNIMAS), Kota Samarahan 94300, Sarawak, Malaysia; 4Chemistry Department, College of Science & Mathematics, Mindanao State University–Iligan Institute of Technology, Iligan City 9200, Lanao del Norte, Philippines; 5Department of Chemistry, College of Natural Sciences and Mathematics, Mindanao State University–Main Campus, Marawi City 9700, Lanao del Sur, Philippines; 6Department of Pathology and Microbiology, School of Medicine, Lusaka P.O. Box 50110, Zambia

**Keywords:** diabetes mellitus, MMPs, wound healing, naringin, eicosane, octacosane, antioxidants

## Abstract

1. Diabetic chronic wounds, mainly foot ulcers, constitute one of the most common complications of poorly managed diabetes mellitus. The most typical reasons are insufficient glycemic management, latent neuropathy, peripheral vascular disease, and neglected foot care. In addition, it is a common cause of foot osteomyelitis and amputation of the lower extremities. Patients are admitted in larger numbers attributable to chronic wounds compared to any other diabetic disease. In the United States, diabetes is currently the most common cause of non-traumatic amputations. Approximately five percent of diabetics develop foot ulcers, and one percent require amputation. Therefore, it is necessary to identify sources of lead with wound-healing properties. Redox imbalance due to excessive oxidative stress is one of the causes for the development of diabetic wounds. Antioxidants have been shown to decrease the progression of diabetic neuropathy by scavenging ROS, regenerating endogenous and exogenous antioxidants, and reversing redox imbalance. Matrix metalloproteinases (MMPs) play vital roles in numerous phases of the wound healing process. Antioxidant and fibroblast cell migration activity of *Marantodes pumilum* (MP) crude extract has previously been reported. Through their antioxidant, epithelialization, collagen synthesis, and fibroblast migration activities, the authors hypothesise that naringin, eicosane and octacosane identified in the MP extract may have wound-healing properties. 2. The present study aims to identify the bioactive components present in the dichloromethane (DCM) extract of *M. pumilum* and evaluate their antioxidant and wound healing activity. Bioactive components were identified using LCMS, HPTLC and GCMS. Excision wound on STZ-induced diabetic rat model, human dermal fibroblast (HDF) cell line and colorimetric antioxidant assays were used to evaluate wound healing and antioxidant activities, respectively. Molecular docking and pkCMS software would be utilised to predict binding energy and affinity, as well as ADME parameters. 3. Naringin (NAR), eicosane (EIC), and octacosane (OCT) present in MP displayed antioxidant action and wound excision closure. Histological examination HDF cell line demonstrates epithelialization, collagen production, fibroblast migration, polymorphonuclear leukocyte migration (PNML), and fibroblast movement. The results of molecular docking indicate a substantial attraction and contact between MMPs. pkCMS prediction indicates inadequate blood-brain barrier permeability, low toxicity, and absence of hepatotoxicity. 4. Wound healing properties of (NEO) naringin, eicosane and octacosane may be the result of their antioxidant properties and possible interactions with MMP.

## 1. Introduction

Diabetes mellitus (DM) is a chronic metabolic disorder that is becoming increasingly prevalent in many developed countries including Malaysia. Diabetic patients with badly controlled glucose levels are highly likely to develop diabetic foot ulcer (DFU) with an incidence rate of 19–34% and DFUs have a tendency to get infected due to the ulcers’ position under the foot [1]. It has been estimated that the global DFU treatment market alone would increase from 7.03 billion USD to 11.05 billion USD by 2027. The most typical reasons are insufficient glycemic management, latent neuropathy, peripheral vascular disease and neglected foot care. In addition, it is a common cause of foot osteomyelitis and amputation of the lower extremities. Patients are admitted in larger numbers attributable to chronic wounds compared to any other diabetic disease. In the United States, diabetes is currently the most common cause of non-traumatic amputations. Approximately five percent of diabetics develop foot ulcers, and one percent require amputation [2]. Hyperglycaemia-induced wounds persist in the inflammatory phase and impair wound closure through the disruption of protein synthesis, migration and proliferation of keratinocytes and fibroblasts [3], thereby impeding the formation of granulation tissue and reducing wound tensile strength.

Oxidative stress greatly contributes to the development of chronic wounds as excessive reactive oxygen species (ROS) generated from the activation of several biochemical pathways including the AGE/RAGE, polyol and hexosamine pathways which induce high oxidative stress that increases advanced glycation end products (AGEs). AGEs have been reported to impair wound contraction and prolong inflammation that hampers ECM proliferation [4]. High level of MMPs, especially MMP-9 leads to the development of non-healing chronic diabetic wounds. Therefore, antioxidants have been widely studied and applied as an effective treatment strategy in diabetic wound treatment and management.

Wound healing is a complex yet organized biological process that restores the anatomic integrity of the skin upon any form of injury. The healing process involves the interaction of various intracellular and extracellular processes which are mostly regulated by a group of enzymes known as the matrix metalloproteinases (MMPs). MMPs are a family of endopeptidases that are involved in releasing growth factors from the ECM, cleavage of growth factor receptors from the cell surface, and ectodomain shedding of adhesion molecules from membranal proteins of the cell surface [5].

The hallmark of the proliferative stage in the wound healing process is the rebuilding of new granulation tissue and ECM synthesis. The organization and maintenance of the ECM are highly dependent on various intracellular and extracellular processes which are mostly regulated by a group of enzymes known as the matrix metalloproteinases (MMPs). Four distinct classes of enzymes are associated with wound healing: collagenases, gelatinases, stromelysins and membrane-type metalloproteinases. Although these enzymes play a pivotal role in cell migration and tissue remodelling, an imbalance of these MMPs leads to excessive degradation which has been linked with the nonhealing nature of diabetic ulcers [6]. As such, MMP inhibitors (MMPIs) are essential in ensuring that the MMP activity is balanced throughout the wound healing process. Many naturally occurring and synthetic MMPIs have been explored for wound healing. Some examples of natural MMP inhibitors include curcumin, resveratrol, theaflavin and catechin derivatives [7].

However, with the evolution of technology such as artificial intelligence, molecular docking and in silico studies, the search for effective natural compounds has become more productive and efficient. *Marantodes pumilum* var. alata, popularly known as Kacip Fatimah, is a Malaysian herb that has been widely used in traditional medicine by women to facilitate post-partum recovery, relieve menstrual problems and treat dysentery. Previous studies have reported *M. pumilum* to possess antibacterial, anti-inflammatory, antioxidant and xanthine oxidase inhibitory properties [8,9,10].

In this study, naringin (NAR), eicosane (EIC) and octacosane (OCT) identified and isolated from the DCM extract of *M. pumilum*, were evaluated for their molecular wound healing and antioxidant activity using in vitro, in vivo and in silico approaches.

## 2. Results

### 2.1. Detection and Identification of Tested Compounds

Naringin, a flavonoid commonly found in berries, was identified from Fraction E (SNP E) of M. pumilum’s DCM extract using high-performance targeted liquid chromatography and liquid chromatography-mass spectrometry with multiple reaction monitoring. On the other hand, eicosane and octacosane were identified from Fraction A (SNP A) of the DCM extract using gas chromatography-mass spectrometry. The results of each identification method are further explained in Section 2.1.1, Section 2.1.2, Section 2.1.3

#### 2.1.1. High-Performance Thin Layer Chromatography (HPTLC)

The HPTLC results showed that Naringin, a flavanone-7-O-glycoside was present in SNP E (Appendix A) and gave a good peak resolution in the analysis of the bioactive constituents present in this sample (Appendix A). Linearity gave an R^2^ value of 0.998. Recovery was able to give CV% less than 2%. Precision gave RSD less than 2% (Appendix A). The proposed HPTLC method for the analysis of SNP Efrom the partially purified leaf of the dichloromethane (DCM) extract of M. pumilum reported here is simple, sensitive, economic, and suitable for rapid routine quality control analysis and quantification of Naringin in herbal drug preparation and may be useful for standardization purposes.

#### 2.1.2. Liquid Chromatography-Mass Spectrometry (LC-MS)

This study used radio-frequency electric fields via the LTQ MS system (linear quadruple trap) to form a trap [11]. Multiple reaction monitoring (MRM) was used to selectively detect target analytes in the sample. Appendix A shows the peaks of standard naringin and catechin whereas Appendix A shows the peaks detected from a sample of SNP E. Fraction E’s major bio-constituent corresponded to Naringin (peak: 4.0 min; 1.18 × 10^4^ cps). Catechin (peak: 3.53 min) was also compared with Fraction E and it was found in trace amounts (Appendix A).

#### 2.1.3. Gas Chromatography-Mass Spectrometry (GC-MS)

The GC/MS analysis of SNP A from the partial purification of M. pumilum is shown in Appendix A. A total of 29 compounds were eluded and the major compounds identified were phthalic esters; Bis(2-propylpentyl) phthalate and Bis(2-propylpentyl) phthalate (RT: 20.65 min, 17.8%), branched alkane hydrocarbons; Heneicosane, Hexadecane-1-iodo, Octacosane (RT: 34.756 min, 11.72%), Pentadecane, Heptadecane and Heptacosane (RT: 23.142 min, 7.89%), Tricosane and Eicosane (RT: 26.902 min, 5.25%) and terpenes; β-amyrin and Urs-12-en-3-one (RT: 28.361 min, 6.84%). The major constituents sum up to 49.5% of the total constituents (Appendix A).

### 2.2. Free Radical Scavenging Studies

The IC_50_ values of the samples against the standard (STD) were tabulated in Table 1. The DPPH radical scavenging activity of NAR was non-significantly different to ascorbic acid with values of 46.3 ± 0.001 µg/mL and 44.3 ± 0.002 µg/mL, indicating the NAR had a similar effect to ascorbic acid in scavenging DPPH radicals. EIC showed a lower IC_50_ value (56.0 ± 0.001 µg/mL) than OCT which yielded a value of 61.7 ± 0.002 µg/mL. However, the NO radical scavenging assay showed that OCT yielded a lower IC_50_ value (123.0 ± 0.001 µg/mL) against EIC (227.0 ± 0.002 µg/mL). The NO assay showed that NAR had a stronger effect than STD with IC_50_ values of 115.3 ± 0.001 µg/mL over 123.3 ± 0.002 µg/mL.

For the hydroxyl and superoxide radical scavenging assays, EIC showed a lower IC_50_ value than OCT with values of 237.3 ± 0.002 µg/mL and 383.0 ± 0.001 µg/mL against 366.3 ± 0.002 µg/mL and 493.0 ± 0.001 µg/mL, respectively. This proved that EIC displayed a better scavenging effect against hydroxyl and superoxide radicals than OCT. The hydroxyl and superoxide radical scavenging activity of NAR yielded IC_50_ values higher than STD (366.3 ± 0.002 µg/mL and 452.0 ± 0.001 µg/mL against 275.7 ± 0.001 µg/mL and 291.0 ± 0.001 µg/mL, respectively). This showed that NAR had a lower hydroxyl and superoxide radical scavenging activity than ascorbic acid.

### 2.3. Fibroblast Migration Studies

The total migration distance of the fibroblasts in the treated and untreated normal and insulin-resistant human dermal fibroblast (HDF) cells are shown in Figure 1a and Figure 1b, respectively. The longest distance of fibroblast migration was observed in both normal and insulin-resistant HDF cells treated with NAR as compared to their respective negative controls with a total distance of 274.67 ± 2.20 µm against 151.06 ± 1.46 µm and 207.85 ± 2.66 µm against 73.81 ± 3.35 µm.

EIC showed better fibroblast migration activity than OCT in the normal HDF cells with a total distance of 292.99 ± 2.91 µm over 257.59 ± 1.57 µm. However, the migration activity of EIC and OCT was similar in the insulin-resistant HDF cells. On the other hand, OCT showed a significant difference in the normal HDF cells as compared to NAR and EIC while all three compounds were non-significant to each other in the insulin-resistant HDF cells.

The fibroblast migration activity of the tested compounds is shown in Figure 2. It was observed that the fibroblasts migrated completely and closed the scratch wound after 24 h in the drug-treated cells as compared to the untreated cells. This indicates that these compounds have a strong ability to facilitate fibroblast migration and could potentially accelerate the wound-healing process.

### 2.4. In Vivo Studies

The in vivo studies comprise physiochemical evaluation of self-formulated Naringin, eicosane and octacosane creams, in vivo wound healing evaluation, histology analyses, and hydroxyproline and glutathione analyses. These will be elucidated individually in Section 2.4.1, Section 2.4.2, Section 2.4.3, Section 2.4.4.

#### 2.4.1. Physiochemical Evaluation of Self-Formulated Creams

Table 2 shows the physiochemical evaluation of the prepared sample creams. The findings show that the formulation used in preparing the creams is stable and suitable for use as a topical application.

#### 2.4.2. Wound Healing Evaluation

Wounds inflicted on the dorsal surface of the diabetic animals were observed and photographed on days 0, 2, 4, 7, 12 and 15, respectively. Figure 3 shows the progressive wound closure of the different treatment groups on diabetic animals. All wounds were closed by day 15.

Figure 4 and Figure 5 show the wound closure area and percentages of wound closure from the different treatment groups on the above-mentioned days from both normal and diabetic animal groups. Based on the results obtained, EIC showed a significantly higher wound area than POS on day 2 (*p* < 0.05). On the other hand, OCT showed a significant wound closure percentage compared to NEG (*p* < 0.0001). NAR, EIC and OCT significantly reduced the wound area on day 15 when compared to the POS group (*p* < 0.01 for NAR vs. POS, *p* < 0.05 for EIC/OCT vs. POS). In the percentages of wound closure, NEG was significantly higher among all groups on day 2 (*p* < 0.05) while EIC had the lowest percentage of wound closure. NAR showed significantly higher wound closure on the day against NEG (*p* < 0.05), EIC (*p* < 0.0001) and OCT (*p* < 0.05).

#### 2.4.3. Histology Analysis

Histological examination of skin tissue from the NEG group demonstrated inflammatory infiltration on days 2 and 4 whereas ectasia vessels with oedema were seen on day 4 (Figure 6). The fibroblasts were also inorganized in the NEG group. On day 7, the NEG group showed a thickening of the epithelial layer. Similarly, epithelialization was observed on day 7 in the POS group.

In the drug-treated groups, naringin was able to complete the epithelial closure by day 7 and there was no oedema observed. The tissue also showed signs of well-defined and structured granulation tissues with matured collagen bundles. Eicosane, on the other hand, showed less defined granulation tissue but showed better maturation of collagen bundles than the NAR group as seen on day 12 (Figure 7). Lastly, octacosane showed delayed epithelial migration with the cut edges still prevalent (<50% cell migration), with a high concentration of inflammatory cells and slight oedema on day 2. On day 12, superficial perivascular lymphocytic infiltration with eosinophils was observed in the OCT group.

The semi-quantitative parameters of each excised wound of the diabetic group on days 2, 4, 7, 12 and 15 were plotted as shown in Figure 8A–C. From these figures, NAR wounds showed a significant difference in each parameter observed against the NEG wound. The increase in epithelialization and fibroblast proliferation of the experimental group shows that NAR has a strong potential in accelerating the wound healing process of diabetic wounds. This is also supported by the decrease of PMNLs over the treatment period, indicating the inflammation of these wounds reduced gradually over a period of 15 days.

It is also evident that EIC and OCT showed strong potential as wound healing agents because a gradual increase in epithelialization and fibroblast proliferation was observed over the treatment period. This is further supported by the decrease in PMNL cells in the excised wounds over the same period of time. EIC and OCT showed no significant difference among the experimental groups on each observed day, indicating that these two compounds have a similar therapeutic effect in treating diabetic wounds.

#### 2.4.4. Hydroxyproline and Glutathione Levels

Day 12 was used in the evaluation of hydroxyproline activity (Table 3). This study was conducted on the normal animal group and animals that had been in a diabetic state for a period of 30 days. There were significantly lower (*p* < 0.05) glutathione enzyme levels in the normal non-wounded skin and wounded skin versus the diabetic groups. The POS group showed a significant increase in GSH levels in the normal group (*p* < 0.001). OCT significantly increased GSH levels (*p* < 0.0001) while EIC decreased the GSH levels in the normal group. NAR increased GSH levels in both normal and diabetic skin tissues.

The hydroxyproline levels did not show a significant difference on the unwounded skin but were significantly elevated in the diabetic animal group on day 12 (*p* < 0.05). POS did not show a significant difference in all groups in comparison to the NEG control. OCT was able to significantly increase the hydroxyproline levels in the normal group (*p* < 0.01) but decreased in the diabetic group (*p* < 0.01). EIC and NAR showed a significant decrease in hydroxyproline levels in the diabetic animal group (*p* < 0.001 and *p* < 0.05, respectively).

### 2.5. Molecular Docking Analysis

In the docking results, the binding conformation with the least-free energy of the binding reflects the strongest binding affinity. All three compounds showed relatively strong binding affinity against the collagenase families as compared to their control ligand (Table 4). However, as a result of the non-polar characteristics of EIC and OCT, only hydrophobic interactions were known to contribute to the increase in binding strength, leading to the reduction of the dissociation constant (kD) values. Conversely, hydrogen bonding interactions were found present in NAR against the collagenase family as the result of its polar characteristics. Both collagenase 1 and 2 (COL1 and COL2) were found to form a total of four hydrogen bonds via Thr241, His218, Asn180, and Ala182 (Figure 9(Ai)) and His207, Pro217, Leu214, and Ala163 (Figure 9(Bi)), respectively. In collagenase 3 (COL3), the number of hydrogen bonds that were present in the binding interactions was greater than in both COL1 and COL2 in which seven of them were formed with Leu185, Ala186, Glu223 and Tyr244 (Figure 9(Ci)).

The predicted kD values for the docked conformation were tabulated in Table 5. Due to the non-polar characteristic of EIC and OCT, the kD value is lesser than that of NAR towards the collagenase family. The binding affinity of both non-polar compounds was stabilized and further strengthened by the hydrophobic surface at the binding site of the collagenase family (Figure 9A–C(ii,iii)). The kD value of NAR is higher although the interaction was present with several hydrogen bonds with the collagenase family. This suggests that hydrophobic interaction might be important in stabilizing the binding interaction.

### 2.6. In Silico Study Analysis

Using the pkCMS web platform, the in silico transdermal absorption was determined using Caco-2 cell models (log P_app_) and skin permeability (log K_p_). In addition to this, the results on the neurotoxicity of the tested compounds were obtained on the basis of the parameters of blood-brain barrier permeability (log BB) and the penetration of the central nervous system (log PS). The toxicity from the parameters of the maximum tolerated doses of the substances, their hepatotoxicity and skin sensitization. Table 6 shows the results obtained from the pkCMS and SwissTargetPrediction software.

According to the model used in the pkCMS software, if the log P_app_ is more than 0.9, it means the compound expresses high Caco-2 permeability. For the tested substances, the log P_app_ value above 0.9 was shown by EIC and OCT, while NAR showed a value of −0.068, indicating that NAR is impermeable to Caco-2 cells. On the other hand, the skin permeability for all tested substances can be predicted to be poor because they have log K_p_ values lower than −2.5.

For NAR, the log BB parameter (−2.034) is clearly lower than that of EIC and OCT (1.03 and 1.166, respectively), which means that it shows poor permeability across the blood-brain barrier. In contrast to this, EIC and OCT showed log BB values above 0.3 and can be considered substances that can be distributed to the brain. NAR with a log PS parameter value of −4.911 can also be considered as unable to penetrate the central nervous system. Substances with a log PS value higher than −2 are considered to penetrate the central nervous system, such as EIC and OCT. Lastly, low toxicity (max. tolerated dose < 0.477) and no hepatotoxicity were determined for all tested substances.

The structures of Naringin (NAR), Eicosane (EIC) and Octacosane (OCT) are presented in Figure 10 whereas Figure 11 displays the target classes of these compounds, respectively.

## 3. Discussion

The prevalence of diabetes mellitus (DM) and its consequences are increasing worldwide, placing a significant burden on individuals and healthcare systems [12]. The effects of diabetic wounds on disability, morbidity and mortality are considerable. The treatment of diabetic wounds remains a significant concern for the medical system. Reports indicate that minimal amounts of reactive oxygen species (ROS) are essential to prevent external damage, but its overproduction leads to the formation of diabetic wounds [13]. Redox imbalance is a leading cause of non-healing diabetic wounds [14] due to an excess of reactive oxygen species (ROS) in tissues and a loss in antioxidant capacity. Long-term type 2 diabetes is associated with severe decreases in antioxidant enzyme activity [15]. Glutathione (GSH) plays an important role in maintaining the redox status of the cell. Reduced GSH levels have been noticed in diabetic wounds due to the disturbance in the redox state of the cell that causes metabolic disturbances as seen in diabetic patients [16]. Some researchers have hypothesized that a hyperglycemic state directly impacts the function of keratinocytes and fibroblasts, causing alterations in protein synthesis, proliferation, and migration, decreased antimicrobial peptide production and increased oxidative stress [17,18,19,20].

Matrix metalloproteinases (MMPs) are zinc-dependent proteolytic enzymes implicated in collagen degradation, cellular interactions and cell-matrix interactions via modulating the levels of cytokines, growth hormones and various biological fragments concealed in ECM [21,22,23,24,25,26]. Imbalance and dysregulation of these enzymes lead to excessive degradation, which relates to non-healing wound processes and the conversion of acute wounds into chronic wounds. Elevated MMPs cause decreased expression of MMP tissue inhibitors (TIMPs). TIMPs are found to be reduced in diabetic chronic wounds which aggravates the problem [27,28,29,30]. The imbalance between MMP and TIMP levels is a major cause of hyperglycemia, hyperlipidemia and hypertension in diabetic ulcers and chronic wounds [31,32]. The overexpression of MMP-13 and MMP-9 is linked to a high concentration of glucose at the wound site [33].

Elevated levels of matrix metalloproteinases (MMPs) have been identified in these wounds because of oxidative stress and glycation end products, which may lead to diabetic peripheral artery disease [34,35]. MMPs are primarily responsible for proteolytic degradation, which has a significant impact on the chronic wound healing process. The deterioration of extracellular matrix (ECM) induced by MMPs, specifically MMP-1, -2, and -9, renders these diabetic wounds worse [36]. In chronic wounds, the mismatch between the level of ECM breakdown and repair is a crucial component causing a delay in the wound healing process. Disproportionate availability of cytokines and other growth factors are needed for wound healing [37,38]. In the absence of TIMP, MMP-1, -8 and -9 are reported to be upregulated in venous wounds [39,40]. To ensure the balance of MMPs, it would require suppressing the effect of MMP-2 and MMP-9 along with upregulating the expression of matrix metalloproteinase inhibitors such as TIMP-1 and TIMP-2 [41]. A second target would be to boost MMP-8 while inhibiting MMP-9. MMP-8 has been demonstrated to accelerate the healing of diabetic wounds in the absence of MMP-9 and vice versa, thus necessitating the inhibition of MMP-9 without affecting MMP-8. Another target is to cleave the MMP-3 zymogen, which is required for MMP-9 activation [42,43,44,45].

In a diabetic rat wound model, the administration of naringin, eicosane and octacosane promoted faster wound healing. Naringin, eicosane and octacosane may promote wound healing due to their potent free radical scavenging, hydroxyproline and glutathione action (antioxidant) as shown in Table 1 and Table 3. Multiple research projects have reported a link between ROS and the development of diabetic wounds. Antioxidants prevent the evolution of diabetic neuropathy by scavenging reactive oxygen species (ROS), regenerating endogenous and exogenous antioxidants, renovating oxidised proteins, inhibiting NF-kB and regulating gene transcription. Antioxidants are commonly employed in clinical settings to treat diabetic microangiopathy because a balanced redox state is likely essential for rapid repair [46,47,48].

Naringin is a water-soluble flavonoid that is present in many citrus fruits whereas eicosane and octacosane are long-chain hydrocarbons that are naturally found in plants such as *Taraxacum officinale*, *Hypericum hircinum* and *Acacia nilotica* [49]. Flavonoids are secondary metabolites that are abundant in various plants with over thousands of individual flavonoid compounds identified up to date. The antioxidant action of flavonoids is frequently studied due to their capability as potent free radical scavengers. Naringin is a flavanone glycoside containing naringenin, a well-studied flavonoid for its antioxidant activity. Flavonoids generally have a molecular structure of a C6-C3-C6 carbon skeleton that comprises two 6-carbon benzene rings linked by a 3-carbon heterocyclic ring [50]. The free radical scavenging ability of naringin is strong due to the location and number of -OH groups and the conjugation and resonance effects between these groups [51]. Naringin has been reported to be effective in reducing oxidative stress in both in vitro and in vivo models [52,53,54,55].

Hydrocarbons, on the other hand, have not been extensively studied for their antioxidant activity. The findings of this study showed that eicosane and octacosane exhibited moderate antioxidant activity as compared to naringin. The reduced antioxidant activity of hydrocarbons is due to the lack of hydroxyl groups as they are mainly composed of carbons and hydrogens. However, a study by Farzaliyev (2012) showed that the antioxidant activity of hydrocarbons can be greatly improved when combined with functional groups that react with peroxide radicals such as phenols and aromatic amines and additives that decompose hydroperoxides such as sulphides [56].

Epidermal growth factor (EGF) administration improves wound healing because of its alleviation of oxidative stress [57]. Biogenic AgNPs synthesized from *Brevibacillus brevis* KN8 could inhibit the overexpression of MMP-2 and MMP-9 in granulation tissues and accelerate wound healing in diabetic mice beyond the antimicrobial activity [58]. Ferulic acid (FA) is a natural antioxidant derived from fruits and vegetables that inhibited lipid peroxidation and increased the expression of catalase, superoxide dismutase, glutathione, nitric oxide and serum zinc and copper, which probably improves the healing process in diabetic ulcers [59]. Syringic acid treatment also promotes migration and proliferation to improve wound healing [60]. The fusion protein decreased serum proinflammatory cytokines such as IL-6, TNF-α, expression of cyclooxygenase-2, and increased activities of antioxidant enzymes including superoxide dismutase, glutathione peroxidase and catalase, and it also increased proangiogenic cytokine levels including VEGF, intercellular adhesion molecule, and expression of VEGF, FGF-2, p-ERK and p-Akt protein in granulation in diabetic rats, which significantly accelerated the diabetic wound healing [61]. Chlorogenic acid, a dietary antioxidant that could accelerate wound healing, enhances hydroxyproline content, decreases malondialdehyde/nitric oxide and elevates the level of reduced glutathione in the wound bed [62].

The fibroblast migration and re-epithelialization activities of naringin, eicosane and octacosane were similar to each other as observed in Figure 1 and Figure 2 (in vitro) and Figure 3, Figure 6, Figure 7 and Figure 8 (in vivo). Fibroblasts play an important role in the proliferative phase of the wound healing process and are involved in ECM synthesis along with other growth factors such as transforming growth factor (TGF-β1) and basic fibroblast growth factor (b-FGF). The multiplication of skin fibroblasts is essential for the earliest stages of wound healing [63]. The effects of naringin, eicosane and octacosane on epithelialization, fibroblast migration, and polymorphonuclear leukocyte migration (PMNL) are depicted in Figure 8. Figure 1 and Figure 2 show that insulin-resistant HDF cells exhibited enhanced fibroblast migration. Epithelialization, fibroblast movement and polymorphonuclear leukocyte migration (PNML) were detected in the naringin, eicosane and octacosane-treated group, possibly because of an increase in collagen synthesis, fibroblast proliferation and tumour necrosis factor-alpha (TNF-) upregulation or by activating the collagen VI expressions and transforming growth factor-beta 1 (TGF-1) as well as its antioxidant capacity [64,65].

These substances caused the cohesion of collagen fibres in the dermis and made connective tissue cells visible in the fibres’ labia. Cellular events for wound repair [66] include angiogenesis, cell migration and proliferation, creation of granulation tissue, collagen synthesis, and re-epithelialization. Consistent with previous studies, treatment of animals with naringin, eicosane and octacosane resulted in increased wound capillary density, re-epithelialization, fibroblast proliferation, and collagen synthesis [67,68]. The interleukins IL-1, IL-6 and TNF- enhance the inflammatory response; collagen production is mediated by FGF-b, IGF and TGF-; and angiogenesis is stimulated by FGF-B, VEGF components A, TGF- and HIF-1 [69,70].

Wound healing is a multicellular process that includes fibroblasts, keratinocytes, endothelial cells and inflammatory cells. Haemostasis, the inflammatory phase, the proliferation phase and the remodelling phase comprise four different yet overlapping phases of the healing process. Crosstalk between different groups of molecules, including extracellular matrix (ECM), integrins, growth factors and MMPs, regulates the phases of wound healing. The migration of cells on extracellular matrix (ECM) and modification and destruction of ECM by matrix metalloproteinases (MMPs) are essential components in wound repair [71]. MMPs can be categorised into ‘Clans’ and ‘Families’ based on fold similarity and evolutionary ties, respectively. The MMP class is comprised of eight clans and about forty families. In accordance with the substrate specificity and homology organisation: (1) Collagenases (MMP-1, MMP-8, MMP-13), (2) Gelatinases (MMP-2 and MMP-9), (3) Stromelysin (MMP-3, MMP-10, MMP-12), (4) Matrilysin (MMP-7, MMP-26), (5) Membrane Type (MT) MMPs (MT-MMP-14, -15, -16, -17, -24, -25) and (6) Additional MMPs (MMP-19, -20, -21, -22, -23, -27, -28). In accordance with the MMPs’ organisational structure: (1) the prototypical MMP (type-1 collagenases), (2) Matrilysins are missing the hemopexin domain, (3) Gelatinases have three type II fibronectin domains and (4) MT-MMPs are localised at the cell membrane surface [72,73].

The molecular docking approach assists to model the interaction between a small molecule and a protein at the atomic level, which allows one to characterize the behaviour of small molecules in the binding site of target proteins as well as to elucidate fundamental biochemical processes [27]. The docking process involves two basic steps: prediction of the ligand conformation as well as its position and orientation within these sites (usually referred to as pose) and assessment of the binding affinity. These two steps are related to sampling methods and scoring schemes, respectively, which will be discussed in the theory section. It provides a prediction of the ligand-receptor complex structure, binding energy and affinity using computation methods. The molecular docking result reveals a strong interaction and affinity between the compounds and MMPs. It shows the compounds’ high chance of modulation of the MMPs. Pharmacologically, the higher the binding affinity, the higher the target occupancy and the higher the chance of activation. This result is in consonance with the observed result. There was evidence of collagen synthesis, fibroblast migration and re-epithelization which are evidence of MMP activation.

The molecular docking performed in this study showed the binding energy of naringin was the highest for collagenases I, II and III against the control, followed by octacosane and eicosane. All three drug compounds showed the highest binding energy with collagenase III compared to collagenases I and II. According to Armstrong and Jude (2002), MMP-13 has a unique ability to cleave type I, II and III collagen [74]. A study by Toriseva et al. (2012) reported that MMP-13 is vital in wound healing as it coordinates the growth of wound granulation tissue and modulates the expression of genes involved in inflammation, proteolysis and cell viability [34]. Collagenase-1, 2 and 3 along with stromelysin 1 and gelatinase B play a significant part in the proliferation and re-epithelization phases of wound healing. It also aids in the granulation of tissue to cover the wound with a dense network of blood vessels. These arteries stimulate the release of platelet-derived growth factors, which aid in wound debridement and reorganisation of damaged type I collagen. It affects the release of inflammatory cytokines, including as IGN and TGF-, as well as the synthesis of collagen and fibroblast synapses [74,75,76,77,78,79,80,81]. The vascular endothelial growth factor (VEGF) secreted by macrophages stimulates the migration and proliferation of keratinocytes and endothelial cells, which comprise MMP-1, 2, 9 and 13, and, therefore, play a significant role in wound healing [82,83].

Gelatinases, such as gelatinase A (MMP-2) and gelatinase B (MMP-9), promote the migration of keratinocytes and the healing process. MMP-9 maintains the wound-healing factors such as vascular endothelial growth factor and dermatopontin [84,85,86]. Gelatinases (MMP-2 and MMP-9) function in keratinocyte migration and angiogenesis. Gelatinase A (MMP-2) is secreted by fibroblasts while gelatinase B (MMP-9) and is produced by leukocytes and possibly, keratinocytes. The binding energies of naringin, eicosane and octacosane were relatively stronger with gelatinase B than with gelatinase A. A study by Hingorani et al. (2018) showed that wound healing was delayed in MMP-2/-9 double knockout (DKO) mice as compared to the wild-type group which suggests that this poor response observed in DKO mice could be attributed to the loss of MMP-9 as it is usually expressed at the leading edge of migrating cells that allows these cells to reepithelialise the wounded site [87]. While several studies support the positive role of MMP-9 in wound healing, it is interesting to note the findings from a study by Reiss et al. (2010) that postulated MMP-9 could be a causal factor of chronic wound development as active MMP-9 expression showed a significant decrease in epithelial migration and type IV collagen [88].

Stromelysins are involved in the degradation of the extracellular matrix. The binding energies of all three drug compounds showed the strongest for stromelysin 1 (MMP-3) and stromelysin 2 (MMP-10). Stromelysins 1 and 2 can cleave several types of collagens, proteoglycans, laminin and fibronectin [89]. Among the tested drug compounds, naringin showed the strongest binding energies with stromelysins 1 and 2, followed by eicosane and octacosane. Stromelysin 1 is responsible for the initiation of wound contraction through the migration of fibroblasts, and downregulation of stromelysin 1 is clearly attributed to delayed wound healing as observed in chronic wounds [90]. Stromelysin 2 is primarily involved in epithelial cell migration and its activity is influenced by cytokines, depending on the cell type. Salmela et al. (2004) reported an increase in stromelysin 2-expressing epithelial cells when cytokines TNF-α or IL-1β was added to the culture media [91].

MMP therapy targets a variety of synthetic and natural substances. Numerous broad-spectrum MMP inhibitors (MMPI) have been studied for this purpose, but we require more specific treatments than those currently available [92]. Compounds such as (R, S)-ND-336, antibodies GS-5745, SSDS-3 and REGA-3G12 and atelocollagen type I, when combined with 4-vinyl-benzyl chloride -CD (D3)7/MMP-9-siRNA, reduce MMP-9 activity in diabetic wounds [84,93,94,95,96]. Withaferin A (3-azido Withaferin A) is a naturally occurring steroidal lactone derived from the plant *Withania somnifera*, while Cantharidin is a naturally occurring chemical derived from the plant *Mylabris phalerata*. Celastrol is an additional phytoconstituent of *Tripterygium wilfordii* with anti-proliferative activity. Ginsenoside Rd is derived from *Panax* ginseng leaf, and lycorine is a naturally occurring substance that is widely dispersed in the *Amaryllidaceae* plant family. All these naturally occurring plant extracts modulate the activity of the MMPI [97,98,99,100,101,102]. Naringin is a flavonoid found naturally in citrus fruits [103]. Treatment with naringin decreases the expression of p-ERK and p-JNK, which are molecular markers implicated in the MAPK signalling pathway, hence decreasing the expression of MMP-2 and MMP-9. Gallic acid has also been reported to activate the natural MMP-2 regulator TIMP-1 [100].

The figure below is a schematic diagram of the mechanism of action of naringin, eicosane and octacosane in the in vitro and in vivo models, and the in silico assessment conducted on the MMPs involved in wound healing (Figure 12).

## 4. Materials and Methods

### 4.1. Identification of Tested Compounds

The tested compound naringin was identified using HPTLC and LC-MS/MRM techniques, whereas eicosane and octacosane were identified using GC-MS. Each technique is further explained in Section 4.1.1–4.1.3.

#### 4.1.1. HPTLC Technique

Of the extracts, 20 μL were separately applied (samples and standard) onto the TLC plate with a 6 mm wide band or spotted with an automatic TLC applicator Linomat-V with N2 flow (Camag, Switzerland), 8 mm from the bottom with instruction input defined from win-CATS-V 1.2.3 software. After sample application, the plates were developed in a 10 × 20 cm horizontal Camag twin glass chamber pre-saturated with the mobile phase (10 mL on each side) for 20 min at room temperature (25–27 °C). The mobile phase consisted of MeOH: EA (60:40 *v/v*). Linear ascending development was carried out until the 8 cm mark. The plates were observed after 30 min of air drying under the Camag UV visualizer (366 nm). Derivation was done with β-Aminoethyl-diphenylborate for α and γ-pyrones (Neu’s reagent) spray which was prepared by dissolving 1 g of diphenylborinic acid aminoethylester in 100 mL of 100% MeOH and was then blow dried. Video Scan software in fluorescence mode was used to quantify the plates post-derivation.

#### 4.1.2. LC-MS/MRM Technique

To further confirm that Fraction E contained Naringin, the sample was analysed by LC-MS and MRM techniques. A Hypersil Gold RP C18 column (2.1 mm I.D. × 100 mm; 3 µm) column was used with two solvents: 0.1% aqueous formic acid (FA) and acetonitrile (AC). Separation was conducted with gradient changes from 98% of 0.1% FA and 2% of AC reaching 100% of AC at 30 min. The flow rate used in this experiment was 200 µL/min and 10 µL injection volume was used. For the MS conditions, the samples used LTQ Ion Trap mass spectrometer (Thermo Scientific) with U-HPLC system (Accela) MRM. The spectral m/z from 100–1000 was recorded, and the MSc fragmentation was carried out with 35–40% collision energy. Electrospray ionization conditions were as follows: source accelerating voltage, 4.0 kV and auxiliary gas, 20 arb.

#### 4.1.3. GC-MS Technique

GC-MS was performed by using Agilent equipment (Agilent 5973 GCMS). In this study HP-5MS 5% Phenyl Methyl Silox column with 30 m in length, and 0.25 µm in diameter was used along with 1 mg/mL of the sample dissolved in methanol. The initial temperature of the oven was programmed to 70 °C then increased by 10 °C/min to 300 °C for 6 min. Front injector: injection volume was 2 µL, run time was 29 min with 1-min post-run. The solvent wash draw speed was 300 µL/min. The solvent wash dispense speed was 6000 µL/min, while the Sample Wash draw speed was 300 µL/min, the sample wash dispense speed was 6000 µL/min, the injection dispense speed was 6000 µL/min and the viscosity delay was 7 s. The mode of injection of the back injector was split, with a heater temperature of 250 °C and a pressure of 8.8085 psi. The total flow was 54 mL/min with a septum purge flow of 3 mL/min. The gas saver was kept on with 20 mL/min after 2 min. purge flow and the split vent was 50 mL/min at a 2 min. run time was 29 min with a 1-min post-run. Compounds elucidated from gas chromatography were purchased for further study.

### 4.2. Free Radical Scavenging Studies

#### 4.2.1. 2,2-diphenylpicrylhydrazyl (DPPH) Radical Scavenging Assay

The 2,2-diphenylpicrylhydrazyl (DPPH) radical scavenging activity of naringin, eicosane and octacosane was carried out according to Khan et al. (2012) with slight modifications [104]. For instance, 1 mL of DPPH (7.8 mg DPPH in 100 mL of methanol) was mixed with 1 mL of the sample solutions (50–500 µg/mL). The tubes were left to stand for 30 min in the dark before measuring the absorbance at 517 nm. Ascorbic acid was used as the positive control.

#### 4.2.2. Nitric Oxide (NO) Radical Scavenging Assay

The nitric oxide (NO) radical scavenging assay was conducted according to [105]. Therefore, 1 mL of 10 mM sodium nitroprusside, 0.25 mL phosphate buffer (pH 7.4) and varying concentrations of the samples (50–500 µg/mL) were added to test tubes and incubated at 25 °C for 2 h. After incubation, 0.5 mL of the reaction mixture was transferred into a new test tube where 1 mL of sulfanilic acid reagent (0.33% in 20% glacial acetic acid) was added. The new mixture was then vortexed and allowed to stand for 5 min to complete diazotization. Lastly, 1 mL of naphthylethylene diamine dihydrochloride (0.1%) was added, mixed and left to stand for 30 min. The absorbance was measured at 540 nm against their corresponding blank solutions (all reagents were added except for the samples). Ascorbic acid was used as the positive control.

#### 4.2.3. Hydroxyl Radical (OH^−^) Scavenging Assay

The hydroxyl (OH^−^) radical scavenging assay was performed according to [105]. Therefore, 0.1 mL of deoxyribose (2.8 mM), 100 µL of EDTA (0.1 mM), 100 µL of hydrogen peroxide (1 mM), 100 µL of ascorbate (0.1 mM) and 100 µL of 20 mM potassium phosphate buffer (pH 7.4) were all mixed in a clean test tube. Sample solutions with different concentrations (50–500 µg/mL) were added to the reaction mixture accordingly and incubated for an hour at 37 °C. The reaction was stopped by individually adding 750 µL of 2.8% TCA and 1% TBA in 50 mM NaOH. The solution was boiled for 10 min and then cooled in water to room temperature. The absorbance of the solution was measured at 520 nm. The positive control used for this experiment was ascorbic acid.

#### 4.2.4. Superoxide Radical (SO^−^) Scavenging Assay

The superoxide (SO^−^) radical scavenging assay was conducted according to [106]. Therefore, 50 mM of phosphate buffer (pH 7.6), 20 µg of riboflavin, 12 mM EDTA and 0.03 mg/mL of nitro blue tetrazolium (NBT) solution were added sequentially prior to the addition of samples (50–500 µg/mL). The absorbance was measured at 590 nm after an incubation period of 15 min. The positive control used was ascorbic acid.

### 4.3. Fibroblast Migration Study

#### 4.3.1. HDF Cell Culture and Differentiation

Normal HDF cells were purchased from Lonza. The cells were cultured using DMEM complete growth media and incubated under standard conditions (37 °C and 5% CO_2_). To induce differentiation of the cells, differentiation media containing DMEM with 2% horse serum (Gibco, New Zealand) and 1% antibiotic was added to the cells. The media was changed every 2–3 days for a period of 5 to 7 days.

#### 4.3.2. In Vitro Scratch Wound Assay

The Ibidi culture insert was used in this assay instead of pipette tips to create equal gaps between the cells. Therefore, 70 µL of the cell suspension that contained 30,000 cells/mL were seeded into each well of the Ibidi culture insert in a µ-Dish. The cells were incubated for 24 h at standard conditions for attachment. The insert was removed the next day to check for attachment. The cell layer was then washed with PBS to remove any unattached cells before adding 2 mL of each sample into the µ-Dish. The dish was then placed on the microscope and moved until the gap was observed. Images were captured at the beginning (0 h) and after 4, 8 and 24 h, respectively.

### 4.4. In Vivo Studies

#### 4.4.1. Cream Preparation of Tested Compounds

The creams of naringin, eicosane and octacosane were individually prepared by weighing 10 g of the sample and mixing it in 100 g of readymade aqua cream (DUOPHARMA, Malaysia). The readymade aqua cream contained 8 g emulsifying wax, 12 g of soft white paraffin and 10% *w/w* liquid paraffin and preservatives: methylparaben (0.32 g) and propylparaben (0.24 g). The mixture was thoroughly mixed and stored in the fridge until further use.

#### 4.4.2. Physiochemical Test of Self-Formulated Creams

(i)Physical Evaluation

The creams were evaluated for their appearance according to subjective organoleptic tests of colour, smell and smoothness of the cream’s application onto the skin. The pH was also determined using a digital pH meter (SevenEasyTM pH meter, Mettler Toledo, Switzerland). For this, 1 g of the cream was dissolved in 100 mL of distilled water and stored for 2 h before measuring its pH. The tests were conducted in triplicates.

(ii)Viscosity

Viscosity was measured using a Modular Advanced Rheometer System (HAAKE MARS 60, ThermoFisher Scientific, Bremen, Germany) at a shear rate of 0.08 s^−^^1^. Dye solubility was used to identify the emulsion type, where 5 g of the prepared cream was mixed with water-soluble amaranth dye and dropped on a microscope slide. The slide was covered with a cover slip and observed under the microscope.

(iii)Stability

An accelerated stability test was performed using hot/cold cycles, where 30 g of the cream was subjected to 40 °C/75% RH for 48 h followed by 2–8 °C for 48 h. This was repeated six times. The long-term stability condition was carried out at 30 °C. 75% RH.

#### 4.4.3. Experimental Design

The study was ethically approved under the code UCSI/2016/PATRICK/23-NOV./801-JAN.-2017-DEC.-2018. A total of 36 male Sprague Dawley rats (200–220 g; six weeks of age) were housed with a controlled temperature of 23 ± 2 °C for 12 h light/12 h dark cycle and fed with food pellets and water ad libitum. The rats were left to acclimatize for two weeks before commencing the experiment. The rats were divided into two groups: - Normal and Diabetic groups. Both groups were further subdivided (*n* = 6) into treated (Bepanthen plus cream as the standard while Naringin, eicosane and octacosane creams as the experimental treatment) and non-treated groups (0.9% saline water).

#### 4.4.4. Induction of Streptozotocin (STZ)

Animals in the diabetic group were injected with a single intraperitoneal (IP) dose of STZ (60 mg/kg) dissolved in sterile cold citrate buffer (0.1 M sodium citrate buffer, pH 4.5). The rats were left for 30 days in a hyperglycemic state to induce chronic glucose toxicity. Blood samples were taken intravenously (IV) via the rat tail for glucose analysis (Accu-chek Performa using the glucose oxidase principle) on three occasions (before diabetic induction, 30 days after diabetic induction and before dissection) to ensure the rats are diabetic until the end of the treatment period. Animals with blood glucose levels higher than 11 mmol/L were considered hyperglycemic.

#### 4.4.5. Animal Experimentation

The rats were anaesthetized with Ilium-Xylazil 100, Ketamil (Troy Laboratories, Glendenning, Australia) and distilled water (1:1:5; 0.1 mL/100 g). The fur on the dorsal surface of the rats was shaved using a sterile surgical blade and the shaved areas were sterilized using 70% alcohol swabs before making three excision wounds using a 5 mm biopsy punch. The treatment of these wounds commenced on the following day when the wounds were first cleaned with 0.9% saline water prior to applying respective treatments. The treatment period was conducted daily for 15 days.

#### 4.4.6. Evaluation of Wound Healing Rate

The wounds were assessed every day by placing a sterile PVC plastic paper over the wound area of the rat to draw the outline of the wound. The PVC paper was then observed under a stereoscope microscope (Olympus) coupled with ToupView 3.7 © software to measure the wound area. The wound closures of both normal and diabetic wounds were measured according to the equation below:Wound closure (%) = wound area [(day 0 − day *x*)/day 0] × 100%,(1)
where *x* is the wound measurement of the treatment on days after day 0.

#### 4.4.7. Histology Using H&E Staining Procedure

The animals were sacrificed on days 0, 2, 4, 7, 12 and 15. The excised skin tissue was first fixed in 10% neutral-buffered formalin (NBF) for 24 h at room temperature followed by routine tissue processing: twice in 10% NBF, once in 70% alcohol, once in 80% alcohol, once in 95% alcohol, thrice in absolute alcohol, twice in xylene and twice in paraffin wax. The embedded tissue pieces in wax were sectioned into thin 5µm slices using a manual microtome (Sakura, Japan). These tissue slices were carefully mounted on microscopic slides to undergo staining using the hematoxylin-eosin (H&E) staining procedure. Once the staining was completed, the slides were covered using a cover slip and viewed using the Zeiss software blue edition and AxioVert.A1 Fluorescence Microscope (Carl Zeiss).

#### 4.4.8. Histological Analysis

A semi-quantitative method was used to evaluate the histological structures of the tissues. Three parameters were observed: (1) epithelialization, (2) polymorphonuclear leukocytes (PMNL) and (3) fibroblast proliferation. These parameters were evaluated based on a study by Rahman et al. (2019) which uses a standardized scale of 0 to 4 as shown in Table 7 [107]. Analysis of re-epithelialization was analysed on days 2 and 4 and the thickness of the epithelial layer was assessed on days 7 and 15. PMNL count were assessed on days 2 to 7 while fibroblast proliferation was analysed on days 4–15. Each parameter was assessed at three different points of the slide at a magnification of 10×.

#### 4.4.9. Hydroxyproline and Glutathione Antioxidant Assays

On day 12 (full wound closure), the tissue samples were used to evaluate the hydroxyproline activity. Skin tissue at least 3–4 mm thick were excised for biochemical testing where 10 mg of the tissue was hydrolysed in 100 µL of concentrated NaOH at 120 °C. A hydroxyproline assay kit (Sigma) was used to analyse the collagen content. Oxidised hydroxyproline was detected by its reaction with 4-(dimethylamino) benzaldehyde (DMAB), which yields a product detectable at 560 nm. The recycling system–glutathione assay (Abnova) was used to evaluate either the reduced or total glutathione present in the skin samples. 5,5′-dithibis-(2-nitrobenzoic acid) or DTNB was used to react with glutathione to produce a yellow product detectable at 412 nm.

### 4.5. Molecular Docking of Tested Compounds on Matrix Metalloproteinases Involved in Wound Healing

To investigate the potential binding of the selected compounds, a molecular docking approach was applied using Autodock 4.2. Matrix metalloproteinases (MMPs) which were potentially involved in the wound healing process were selected as protein targets to determine the binding potential for naringin, eicosane and octacosane. The selected proteins are namely stromelysin-1, 2 and 3, matrilysin, collagenase-1, 2 and 3, metalloelastase, metalloproteinase and gelatinase-A and B. The 3D structure of the above-mentioned proteins was retrieved from Protein Data Bank (Appendix A). All the selected structures were subjected to control docking with the co-crystalized inhibitor prior to the docking against naringin, eicosane and octacosane. The binding affinity of docking results was visualized using Visual Molecular Dynamics (VMD) and Ligplot+ for the interaction analysis and the dissociation constant (K_D_) predicted values were obtained from the Prodigy server.

### 4.6. In Silico Study of Drug Compounds

The title substances naringin, eicosane and octacosane were subjected to in silico pharmacokinetic and ADMET profile analyses as well as biological target determination using the pkCMS and SwissTargetPrediction online platforms, respectively [108,109].

### 4.7. Statistical Analysis

Analysis of variance (ANOVA) followed by various post-hoc tests was used to determine the significant differences between the mean data collected from the observations made. The standard deviation (SD) and standard error mean (SEM) values were calculated and represented with the observed data, respectively.

## 5. Conclusions

Naringin, eicosane and octacosane displayed antioxidant and wound closure properties. Histological analysis of the wound and HDF cell line revealed epithelialization, collagen synthesis, fibroblast migration, polymorphonuclear leukocyte migration (PNML), hydroxyproline activity, glutathione activity and fibroblast migration. Molecular docking experiments suggest a substantial binding contact and affinity with MMPs including COL1-3, STRO1-3, matrilysin, metalloelastase, metalloproteinase, GEL A and B. Insufficient blood-brain barrier permeability, minimal toxicity and an absence of hepatotoxicity are predicted by pkCMS ADME software. Naringin, eicosane and octacosane may have wound healing benefits due to their antioxidant characteristics and probable interactions with MMP. These are potential wound healing development leads. Further research is needed to elucidate the MMPs being targeted by naringin, eicosane and octacosane and to ascertain whether they are inhibitors MMPs 2 and 9 and activators of TIMPs. Naringin, eicosane and octacosane will be incorporated into a cream for human trial their effect in wound healing.

## 6. Limitation of Study

Although the rats were of the same age and weight metabolic level may not be same and affect the wound healing speed of individual rats. Even though same dosage of STZ was used for DM induction, their individual response may vary. The specific MMPs targeted by compound was not established yet as molecular docking preliminary forecast of binding modulation.

## Figures and Tables

**Figure 1 molecules-28-01043-f001:**
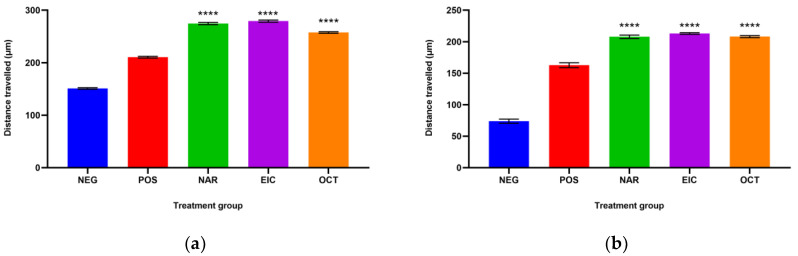
The total distance travelled by (**a**) normal HDF cells and (**b**) insulin-resistant HDF cells with different treatment groups. **** *p* < 0.0001 against the negative control group.

**Figure 2 molecules-28-01043-f002:**
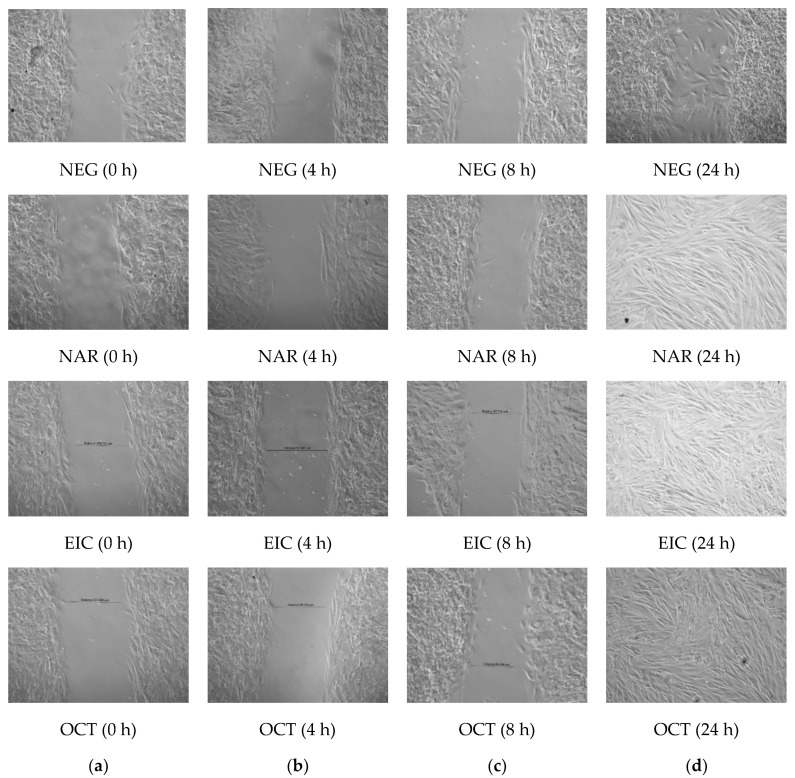
Fibroblast migration of untreated (NEG) vs. treated cells (POS, NAR, EIC and OCT) at (**a**) 0 h (**b**) after 4 h (**c**) after 8 h and (**d**) after 24 h.

**Figure 3 molecules-28-01043-f003:**
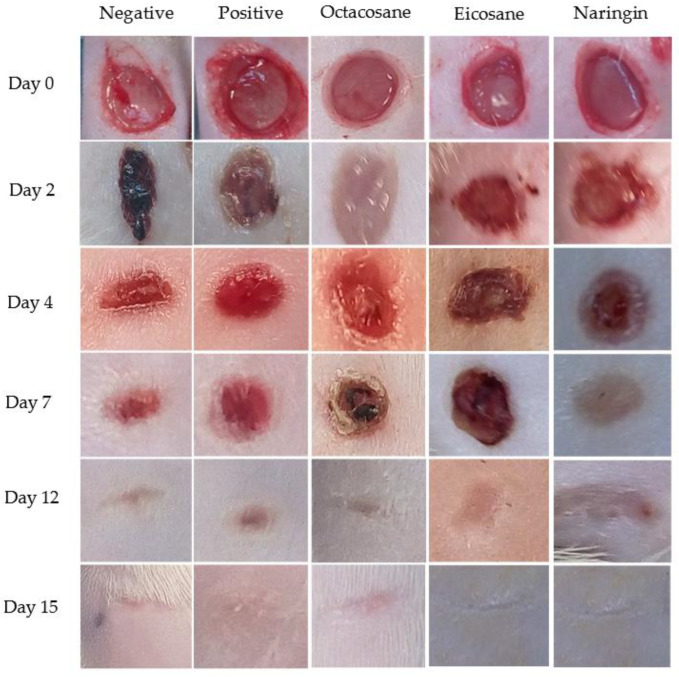
Excised wounds of different treatment groups from the diabetes-induced animal model on different days over a period of 15 days.

**Figure 4 molecules-28-01043-f004:**
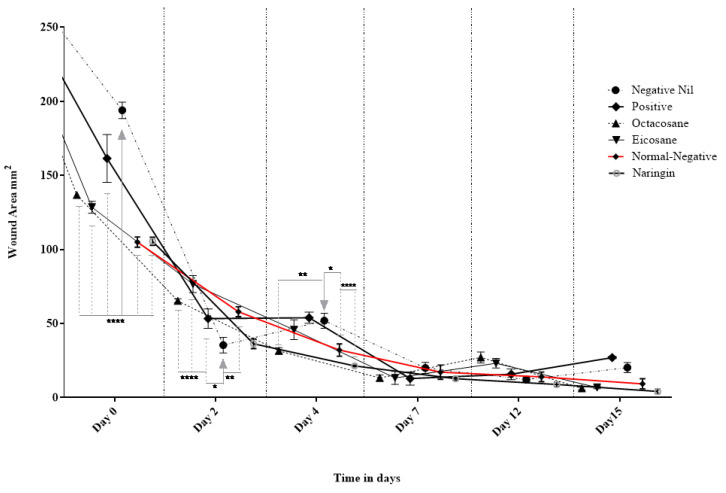
Effect of Naringin, Eicosane and Octacosane treatments on the wound area of diabetes-induced animals (mm^2^). * *p* < 0.05, ** *p* < 0.01, **** *p* < 0.0001.

**Figure 5 molecules-28-01043-f005:**
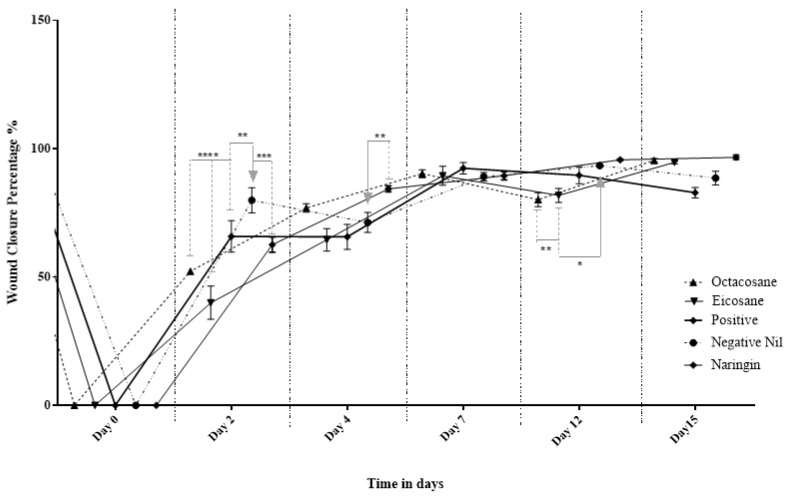
Percentage wound contraction of diabetic wounds from different treatment groups. * *p* < 0.05, ** *p* < 0.01, *** *p* < 0.001, **** *p* < 0.0001.

**Figure 6 molecules-28-01043-f006:**
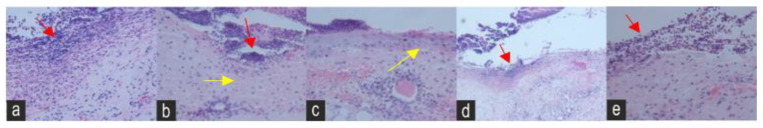
Histopathology sections of the (**a**) negative control (NEG) (**b**) positive control (POS) (**c**) Naringin (NAR) (**d**) Eicosane (EIC) and (**e**) Octacosane (OCT) groups on day 2 under an ×40 magnification. The red arrows indicate PMNL cells; the yellow arrows indicate keratinocyte migration.

**Figure 7 molecules-28-01043-f007:**
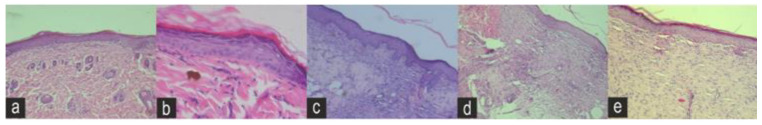
Histopathology sections of the (**a**) negative control (NEG) (**b**) positive control (POS) (**c**) Naringin (NAR) (**d**) Eicosane (EIC) and (**e**) Octacosane (OCT) groups on day 12 under an ×20 magnification.

**Figure 8 molecules-28-01043-f008:**
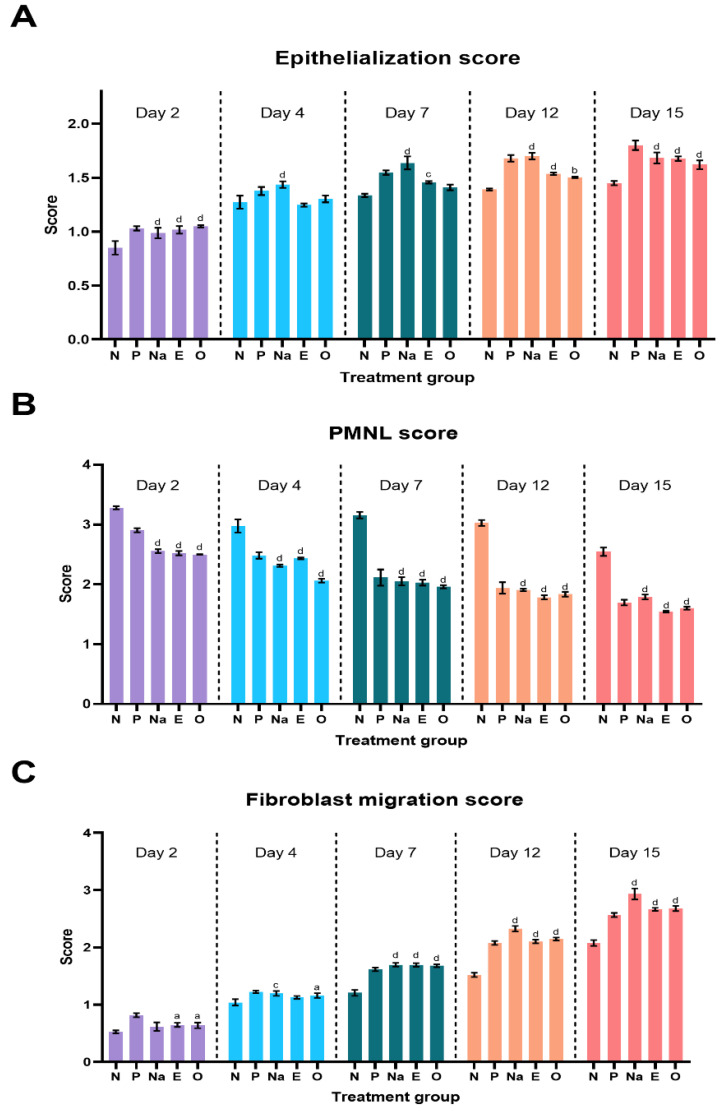
Semi-quantitative parameter analysis of (**A**) epithelialization score (**B**) PMNL score and (**C**) fibroblast migration score. Letters ‘a’ represents *p* < 0.05, ‘b’ for *p* < 0.01, ‘c’ for *p* < 0.001 and ‘d’ for *p* < 0.0001 against the negative group of their respective days.

**Figure 9 molecules-28-01043-f009:**
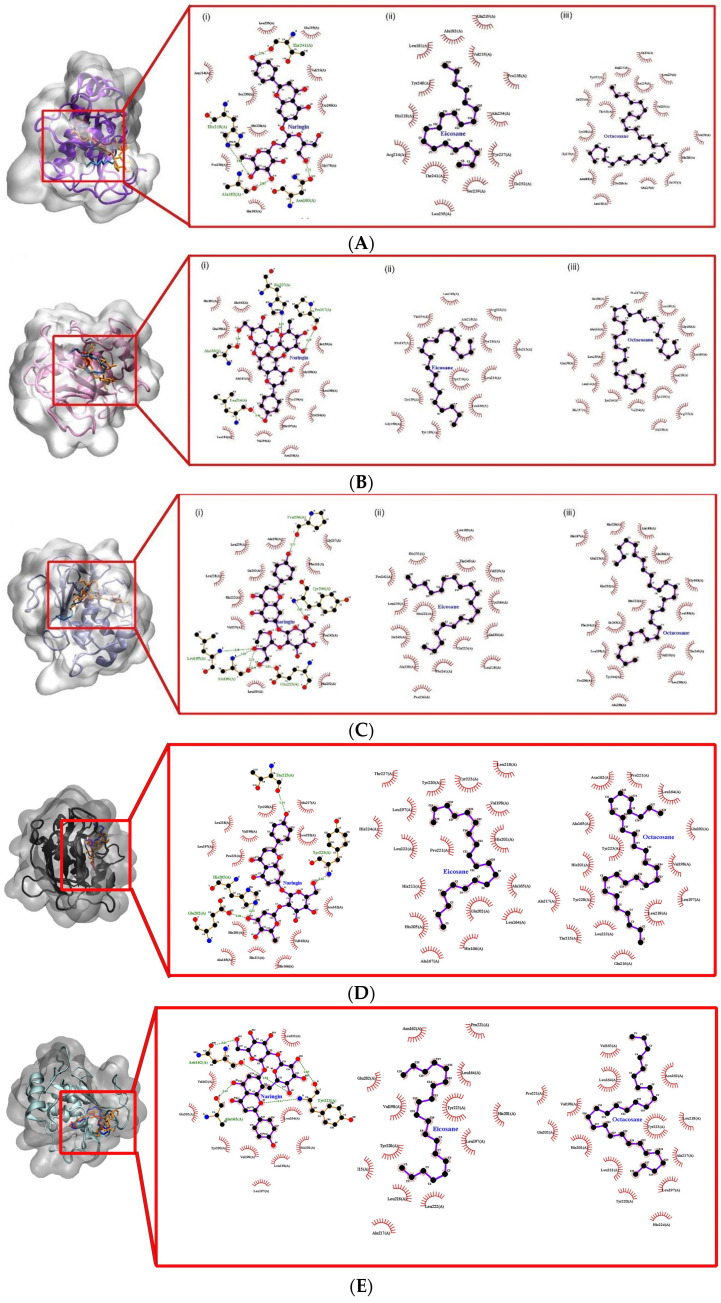
Docked conformation and interaction analysis of (**A**) COL1 (**B**) COL2 and (**C**) COL3 (**D**) STRO1 (**E**) STRO2 (**F**) STRO3 (**G**) Matrilysin (**H**) Metalloelastase (**I**) Metalloproteinase (**J**) GEL A and (**K**) GEL B against NAR (orange—(**i**)), EIC (red—(**ii**)) and OCT (blue—(**iii**)). The compounds were docked at the binding sites of the control ligand/co-crystallized ligand. Hydrophobic interaction is represented as spokes whereas hydrogen bonding interaction is represented with the green dotted line.

**Figure 10 molecules-28-01043-f010:**
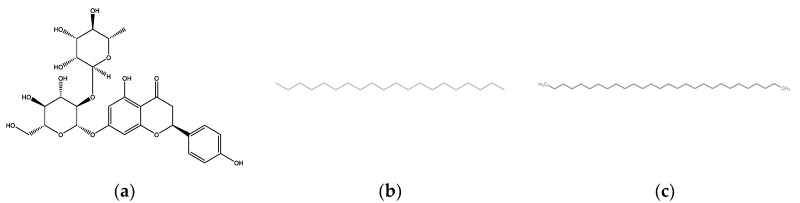
Structure of bioactive compounds (**a**) Naringin (**b**) Eicosane and (**c**) Octacosane.

**Figure 11 molecules-28-01043-f011:**
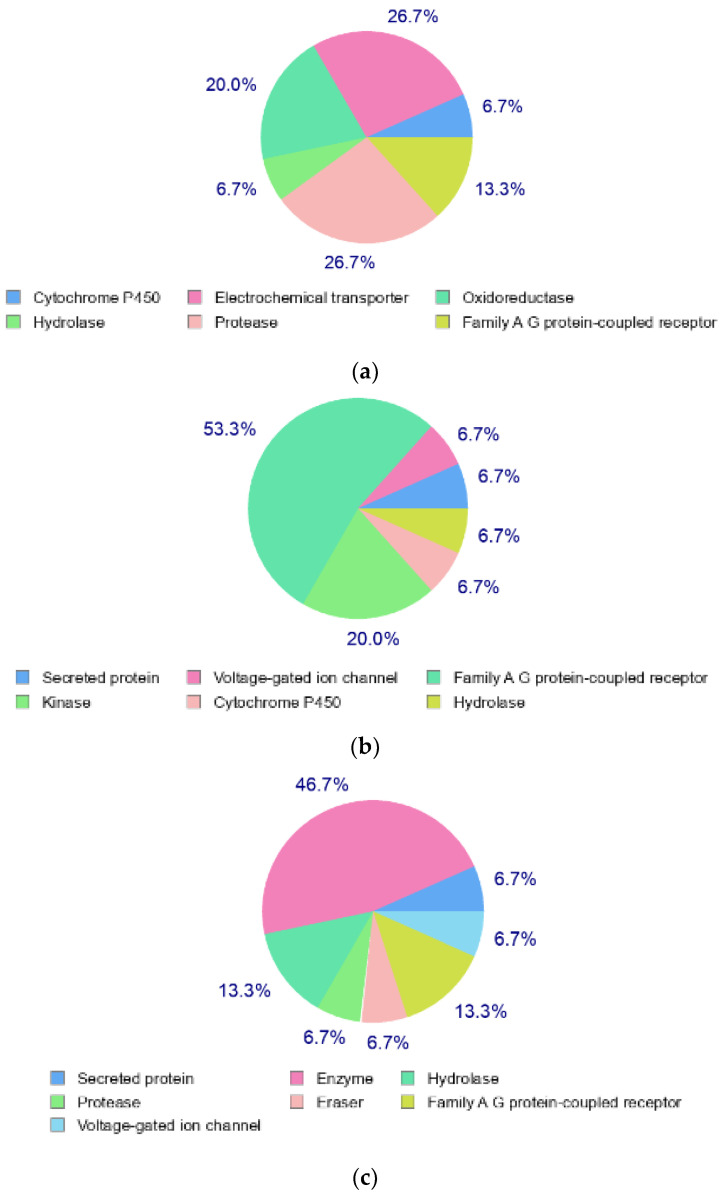
Target classes of compounds (**a**) Naringin (**b**) Eicosane and (**c**) Octacosane.

**Figure 12 molecules-28-01043-f012:**
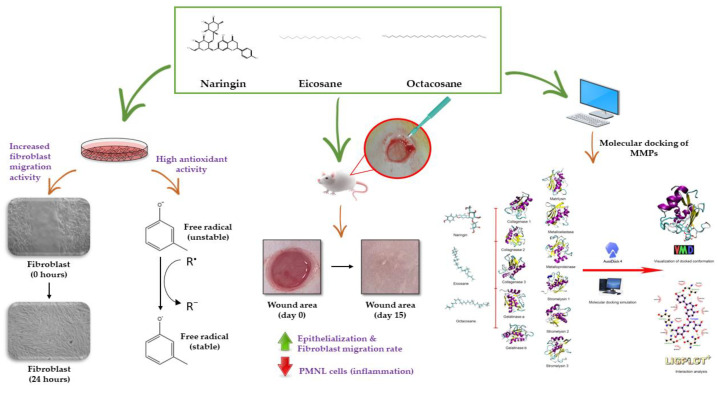
Mechanism of action of naringin, eicosane and octacosane through the in vitro and in vivo models, and the molecular docking of these compounds with the MMPs involved in wound healing.

**Table 1 molecules-28-01043-t001:** IC_50_ values of samples against ascorbic acid (standard).

Assays	IC_50_ (µg/mL)			
	STD	NAR	EIC	OCT
DPPH	44.3 ± 0.002	46.3 ± 0.001	56.0 ± 0.001 *	61.7 ± 0.002 *
NO	123.3 ± 0.002	115.3 ± 0.001	227.0 ± 0.002 *	123.0 ± 0.001
OH^−^	275.7 ± 0.001	336.3 ± 0.002 *	237.3 ± 0.002 *	366.3 ± 0.002 *
SO^−^	291.0 ± 0.001	452.0 ± 0.001 *	383.0 ± 0.001 *	493.0 ± 0.001 *

* *p* < 0.0001 against the standard.

**Table 2 molecules-28-01043-t002:** Physiochemical evaluation of self-formulated sample creams.

Parameters	Sample Creams		
	NAR	EIC	OCT
Appearance	White, non-greasy, odourless and smooth		
pH	5.4 ± 0.1	5.6 ± 0.2	5.3 ± 0.23
Viscosity (poises)	1604 ± 257	1779 ± 28	1756 ± 257
Storage stability	No physiochemical changes observed		

**Table 3 molecules-28-01043-t003:** Hydroxyproline and glutathione activity of different treatments in normal and diabetic groups.

Treatment Group	Hydroxyproline Activity (nM)		Glutathione Activity (nM)	
	Normal	Diabetic	Normal	Diabetic
NEG	11.89 ± 0.271	18.24 ± 1.662 ^a^	1.011 ± 0.001	1.233 ± 0.013 *
POS	13.25 ± 0.12	21.18 ± 1.585 *	1.038 ± 0.011 ^d^	1.462 ± 0.024 *^,c^
NAR	14.34 ± 0.017	13.23 ± 0.102	1.223 ± 0.013 ^d^	1.213 ± 0.013
EIC	12.09 ± 0.317	9.04 ± 0.395 ^c^	0.969 ± 0.011 ^d^	1.008 ± 0.003
OCT	24.67 ± 4.609 ^b^	13.36 ± 0.089 *^,b^	1.097 ± 0.003	1.264 ± 0.013 *^,d^

All values were expressed as mean ± SEM (*n* = 3). Two-way ANOVA with Tukey’s test among groups: * *p* < 0.01. One-way ANOVA with Bonferroni’s test for each parameter: ^a^
*p* < 0.05, ^b^
*p* < 0.01, ^c^
*p* < 0.001, ^d^
*p* < 0.0001 against the negative control group.

**Table 4 molecules-28-01043-t004:** Lowest binding energies of studied compounds with the proteins.

Protein	Free Energy of Binding (Kcal/mol)			
	Control	Naringin	Eicosane	Octacosane
COL1	−10.16	−9.06	−5.99	−6.32
COL2	−6.48	−9.37	−6.42	−5.83
COL3	−7.29	−11.32	−6.66	−7.05
STRO1	−9.75	−8.87	−6.32	−6.36
STRO2	−10.50	−8.54	−6.75	−6.40
STRO3	−16.31	−7.88	−6.27	−4.77
Matrilysin	−2.93	−7.43	−4.66	−4.14
Metalloelastase	−10.86	−9.68	−6.81	−6.77
Metalloproteinase	−2.98	−4.47	−1.49	−0.86
GEL A	−4.66	−7.56	−5.75	−6.51
GEL B	−12.06	−9.33	−8.00	−6.80

**Table 5 molecules-28-01043-t005:** Dissociation constant (kD) of studied compounds against the proteins.

Protein	kD (Kcal/mol)		
	Naringin (NAR)	Eicosane (EIC)	Octacosane (OCT)
COL1	−5.3	−9.6	−10.7
COL2	−5.3	−9.4	−10.6
COL3	−5.3	−9.4	−10.8
STRO1	−5.4	−9.8	−10.7
STRO2	−5.3	−9.3	−10.5
STRO3	−5.4	−9.7	−10.4
Matrilysin	−5.4	−9.6	−10.8
Metalloelastase	−5.3	−9.6	−10.7
Metalloproteinase	−5.3	−8.6	−9.4
GEL A	−5.4	−9.8	−11.3
GEL B	−5.4	−10.0	−11.5

**Table 6 molecules-28-01043-t006:** The selected ADME parameters for the tested compounds.

Parameters Observed	Tested Compounds		
	NAR	EIC	OCT
Water solubility (log mol/L)	−2.891	−8.525	−7.085
Caco-2 permeability (log P_app_ in 10^−6^ cm/s)	−0.68	1.38	1.114
Skin permeability (log K_p_)	−2.735	−2.78	−2.743
BBB permeability (log BB)	−2.034	1.03	1.166
CNS permeability (log PS)	−4.911	−0.937	−0.762
Max. tolerated dose (human) (log mg/kg/day)	0.323	−0.188	−0.284
Hepatotoxicity	No	No	No
Skin sensitization	No	Yes	Yes

**Table 7 molecules-28-01043-t007:** Semi-quantitative histological scoring of wound healing parameters.

Score	Epithelialization	PMNL	Fibroblasts
0	Thickness of cut edges	Absent	Absent
1	Cell migration (<50%)	Mild ST	Mild ST
2	Cell migration (>50%)	Mild DL/GT	Mild GT
3	Bridging the excision	Moderate DL/GT	Moderate GT
4	Keratinization	Marked DL/GT	Marked GT

ST: surrounding tissue; GT: granulation tissue; DL: demarcation line.

## Data Availability

Not applicable.

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
