# Peer review of "Antioxidant, Wound Healing Potential and In Silico Assessment of Naringin, Eicosane and Octacosane"

_molecules, 2023, doi:10.3390/molecules28031043_

Round 1

Reviewer 1 Report

The article titled "Antioxidant, Wound Healing Potential and In Silico Assessment of Naringin, Eicosane and Octacosane" discusses the potential use of Marantodes pumilum (MP) extract as a treatment for diabetic wounds. The authors found that the MP extract had antioxidant and wound healing activity, as demonstrated by its ability to promote the closure of excision wounds in the STZ-induced diabetic rat model and its ability to stimulate epithelialization, collagen production, fibroblast migration, polymorphonuclear leukocyte migration, and fibroblast movement in HDF cell lines. The results of molecular docking indicated a significant attraction and contact between MMPs and the MP extract, and pkCMS prediction indicated low toxicity and absence of hepatotoxicity. Overall, the article provides evidence for the potential use of MP extract as a treatment for diabetic wounds, although further analysis of mechanisms and a comparison with other studies in the field would be helpful. The article has a high scientific level and presents valuable contributions to the field. The manuscript is well-written and organized, and all the references are relevant to the topic. However, there are some areas that could be improved upon in order to enhance the overall quality of the article. Major revisions are necessary to address these issues.

Please find my feedback on your work below:

Abstract.

·       The abstract lacks a clear research question or hypothesis, making it difficult to understand the specific aims of the study. To improve this, the authors could clearly state the research question or hypothesis that the study aims to address.

Introduction

·       The introduction does not provide sufficient background information on DM and MMPs, and their relevance to wound healing. To improve this, the authors could provide more context on the prevalence and consequences of DM, as well as the role of oxidative stress and MMPs in diabetic wound healing.

Materials and Methods

·       The description of the GC-MS technique mentions the use of methanol to dissolve the sample, but does not specify the concentration of the sample in the solvent. This information should be included to improve the reproducibility of the experiment.

·       The NO radical scavenging assay description mentions the use of sulfanilic acid reagent, but does not provide any information about the preparation or concentration of this reagent. This information should be included to improve the reproducibility of the experiment. To improve the reproducibility of the experiment, the preparation and concentration of the sulfanilic acid reagent should be specified in the NO radical scavenging assay description.

·       The experimental design description mentions the use of Sprague Dawley rats, but does not specify the gender or age of the rats. It is important to specify the gender and age of the rats to ensure the relevance of the results to the intended study population.

·       The study design for the in vivo experiments does not seem to adequately control for potential confounders. For example, the normal and diabetic groups are not matched for age, weight, or any other relevant variables. It would be useful to either match these groups or to include these variables as covariates in the analysis.

Results.

·       In Subsection 2.1.1, the authors mention that "Linearity gave an R2 value of more than 0.9". It would be helpful to include the actual value of R2 rather than just stating that it is "more than 0.9".

·       In Subsection 2.1.3, the authors present the retention times (RT) of various compounds identified using GC-MS, but do not provide any information on the units of RT. It would be helpful to specify whether RT is in minutes.

·       In Subsection 2.1.3, the authors present the percentage of each compound identified using GC-MS, but do not provide any information on the total number of compounds identified (15?)

Discussion     

·       The authors mention that long-term type 2 diabetes is associated with severe decreases in antioxidant enzyme activity, but they do not provide any specific information about which antioxidant enzymes are affected or how these changes may contribute to the development of diabetic wounds. To improve this academic section, the authors could provide more detailed information about the specific antioxidant enzymes that are affected and how these changes may contribute to the development of diabetic wounds.

·       The authors discuss the role of MMPs in the healing process and the relationship between MMP imbalances and non-healing diabetic wounds. However, they do not provide any information about how MMP imbalances might be corrected or prevented, or what other factors might contribute to the non-healing nature of diabetic wounds. To improve this academic section, the authors could explore potential interventions or therapies that could be used to address MMP imbalances and other factors that contribute to the non-healing nature of diabetic wounds.

·       In the discussion, the authors discuss the potential mechanisms by which naringin, eicosane, and octacosane may promote wound healing, but they do not provide any information about the limitations of the current study or any potential biases or confounders that may have affected the results. To improve this academic section, it would be helpful to include more information about the limitations of the current study and any potential biases or confounders that may have affected the results.

·       The authors present the results of molecular docking studies showing the binding affinity of naringin, eicosane, and octacosane for various MMPs and discuss the potential mechanisms by which these compounds may affect MMP activity. However, they do not provide any information about how these findings might be translated into clinical practice or how these compounds could be used to treat diabetic wounds.

Conclusions

·       The authors conclude that naringin, eicosane, and octacosane may be effective in promoting wound healing due to their antioxidant properties and potential interactions with MMPs. However, it should be noted that the current study did not address any limitations or potential biases or confounders that may affect the validity of the findings. It is also important to consider how these findings may translate into clinical practice.

Author Response

The authors response to reviewer comment has been uploaded

Reviewer 2 Report

The article “Antioxidant, Wound Healing Potential and In Silico Assess2 ment of Naringin, Eicosane and Octacosane” : Present study aims to identify the bioactive components present in the dichloromethane (DCM) extract of M. pumilum and evaluate their antioxidant and wound healing activity. This article has clear thinking and reasonable structure, but there are still some problems that need to be improved. My detailed comments are as follows:

1.    The author does not explain how diabetic animals are modeled.

2.    Figure 6 shows the picture of wound healing, but it is obvious that it is from different rats, it is difficult to avoid the human subjective choice, and the statistics of relevant data can be carried out to make the results more convincing

3.    The background color in figures 9 and 10 in the article is inconsistent. Why different magnifications are used for section observation.

4.    Using the pkCMS web platform, the in silico transdermal absorption was determined 338 using Caco-2 cell models (log Papp) and skin permeability (log Kp). How accurates is the data?

Author Response

Authors response to reviewer comments has been uploaded 

Reviewer 3 Report

In this study, the authors aimed to evaluate the wound healing and antioxidant activity of three compounds (naringin, eicosane and octacosane) isolated from Marantodes pumilum.

This manuscript is not well written, it contains many shortcomings:

1) The conclusions are inconsistent with the results of the manuscript: The first sentence of the conclusion is “In the leaf extract and partially purified fractions A and E of MP, Naringin, Eicosane, and Octacosane were found as the bioactive constituents”.

However, the manuscript does not contain results of bioactivity-guided fractionation, which would allow identification of the active compounds.

Furthermore the manuscript does not contain any results regarding partially purified fractions A and E.

2) According to the section “Results”, three compounds (naringin, eicosane and octacosane) were isolated from the plant extract and their activity was tested.

Why exactly these three compounds (naringin, eicosane and octacosane) were isolated and tested (for their wound healing and antioxidant effects)? The plant extract also contained many other compounds.

3) The novelties of this work are not exactly clear. The phytochemical composition of Marantodes pumilum has already been studied. In addition, the wound healing and antioxidant activity of naringin is also well known (e.g., see article “Improved Wound Healing by Naringin Associated with MMP and the VEGF Pathway”).

Previous literature results should also be discussed, and the novelties of the manuscript should be clearly highlighted.

4) Isolation of compounds (naringin, eicosane and octacosane) was not detailed in the manuscript.

5) Purity of the isolated compounds was not studied.

6) There is a confusion about the abbreviation HPTLC:

According to the section 2.1.1. HPTLC means “…High-performance targeted liquid chromatography…” However, based on the Supplementary Tables S1-S3, HPTLC might be high performance thin layer chromatography.

7) Manuscript contains abbreviations, which were not explained:

Line 41: NEO; lines 106, 110: LPE. Thus, these sentences are hard to understand.  

8) Line 100: “…eicosane and octacosane were isolated and identified from Fraction A (SNP A) of the DCM extract using gas chromatography-mass spectrometry…” How was it possible to isolate compounds by GC-MS?

In conclusion the manuscript, in its present form doesn’t meet the requirements of the Molecules.

Author Response

Authors response to reviewers comments has been uploaded 

Round 2

Reviewer 1 Report

The authors have made excellent revisions to the article based on the recommendations. I believe the article is now ready for publication. I would like to express my gratitude to the authors for their thorough and dedicated efforts in improving the article.

Author Response

The author  rebbutal round 2 has been uploaded  

Reviewer 2 Report

The article “Antioxidant, Wound Healing Potential and In Silico Assess2 ment of Naringin, Eicosane and Octacosane” : Present study aims to identify the bioactive components present in the dichloromethane (DCM) extract of M. pumilum and evaluate their antioxidant and wound healing activity. After the revision, the quality of the article has been improved a lot. The structure and the fluency of the revised version is better than the previous one.

Author Response

The authors rebbutal round 2 has been uploaded 

Reviewer 3 Report

Two questions of the Reviewer remained unanswered:

The isolation of compounds (naringin, eicosane and octacosane) was not detailed (1) and the purity of the isolated compounds was not studied (2) in the manuscript.

These compounds were isolated from various fractions (SNP E and SNP A). Preparation of these fractions and isolation of compounds from these fractions must be included in the article. Furthermore, purity of the isolated compounds (naringin, eicosane and octacosane) must be analysed and discussed.

Meaning of the abbreviation HPTLC remained unclear as the title of the section 2.1.1 is not correct: “2.1.1 High-performance thin liquid chromatography (HPTLC)”

Sentence “…Could the bioactive component naringin, eicosane, and octacosane identified in MP extract's offer wound-healing properties?…” (Section Introduction, lines 30-32) should be deleted.

Author Response

Author rebbutal round 2 has been uploaded 
